# Susceptibility and infectiousness of SARS-CoV-2 in children versus adults, by variant (wild-type, alpha, delta): A systematic review and meta-analysis of household contact studies

Olalekan A. Uthman[1]*, Frederik Plesner Lyngse[2], Seun Anjorin[1], Barbara Hauer[3], Seran Hakki[4], Diego A. Martinez[5,6], Yang Ge[7], Jakob Jonnerby[8], Cathinka Halle Julin[9], Gary Lin[10], Ajit Lalvani[4], Julika Loss[11], Kieran J. Madon[4], Leonardo Martinez[12], Lisbeth Meyer Næss[8], Kathleen R. Page[5], Diana Prieto[6], Anna Hayman Robertson[8], Ye Shen[13], Juliane Wurm[10], Udo Buchholz[3]

1 Warwick Centre for Global Health, Warwick Medical School, University of Warwick, Coventry, United Kingdom, 2 Department of Economics & Center for Economic Behaviour and Inequality, University of Copenhagen, Copenhagen, Denmark, 3 Department for Infectious Disease Epidemiology, Respiratory Infections Unit, Robert Koch Institute, Berlin, Germany, 4 NIHR Health Protection Research Unit in Respiratory Infections, National Heart and Lung Institute, Imperial College London, London, United Kingdom, 5 Department of Medicine, Johns Hopkins University School of Medicine, Baltimore, MD, United States of America, 6 School of Industrial Engineering, Pontificia Universidad Católica de Valparaíso, Valparaíso, Chile, 7 School of Health Professions, University of Southern Mississippi, Hattiesburg, Mississippi, United States of America, 8 School of Public Health, Imperial College London, London, United Kingdom, 9 Division of Infection Control, Norwegian Institute of Public Health, Oslo, Norway, 10 Department of Emergency Medicine, Johns Hopkins University School of Medicine, Baltimore, MD, United States of America, 11 Department of Epidemiology and Health Monitoring, Robert Koch-Institute, Berlin, Germany, 12 Department of Epidemiology, Boston University School of Public Health, Boston, MA, United States of America, 13 Department of Epidemiology and Biostatistics, College of Public Health, University of Georgia, Athens, GA, United States of America

* olalekan.uthman@warwick.ac.uk

**Data Availability Statement:** All relevant data are within the paper and its Supporting Information files.

## Abstract

### Importance

Understanding the susceptibility and infectiousness of children and adolescents in comparison to adults is important to appreciate their role in the COVID-19 pandemic.

### Objective

To determine SARS-CoV-2 susceptibility and infectiousness of children and adolescents with adults as comparator for three variants (wild-type, alpha, delta) in the household setting. We aimed to identify the effects independent of vaccination or prior infection.

### Data sources

We searched EMBASE, PubMed and medRxiv up to January 2022.

**Funding:** "This research was commissioned by the Robert Koch Institute, Germany. Frederik Plesner Lyngse was supported by the Independent Research Fund Denmark (Grant no. 9061-00035B); Novo Nordisk Foundation (grant no. NNF17OC0026542); the Danish National Research Foundation through its grant (DNRF-134) to the Center for Economic Behavior and Inequality (CEBI) at the University of Copenhagen. The funders had no role in study design, data collection and analysis, decision to publish, or preparation of the manuscript."

**Competing interests:** The authors have declared that no competing interests exist.

### Study selection

Two reviewers independently identified studies providing secondary household attack rates (SAR) for SARS-CoV-2 infection in children (0–9 years), adolescents (10–19 years) or both compared with adults (20 years and older).

### Data extraction and synthesis

Two reviewers independently extracted data, assessed risk of bias and performed a random-effects meta-analysis model.

### Main outcomes and measures

Odds ratio (OR) for SARS-CoV-2 infection comparing children and adolescents with adults stratified by wild-type (ancestral type), alpha, and delta variant, respectively. S*usceptibility* was defined as the secondary attack rate (SAR) among susceptible household contacts irrespective of the age of the index case. *Infectiousness* was defined as the SAR irrespective of the age of household contacts when children/adolescents/adults were the index case.

### Results

Susceptibility analysis: We included 27 studies (308,681 contacts), for delta only one (large) study was available. Compared to adults, children and adolescents were less susceptible to the wild-type and delta, but equally susceptible to alpha. Infectiousness analysis: We included 21 studies (201,199 index cases). Compared to adults, children and adolescents were less infectious when infected with the wild-type and delta. Alpha -related infectiousness remained unclear, 0–9 year old children were at least as infectious as adults. Overall SAR among household contacts varied between the variants.

### Conclusions and relevance

When considering the potential role of children and adolescents, variant-specific susceptibility, infectiousness, age group and overall transmissibility need to be assessed.

## Introduction

Since emergence of SARS-CoV-2, the virus has caused hundreds of millions of COVID-19 cases and millions of deaths [1]. Age-disaggregated data reported to WHO show that children and adolescents are underrepresented, particularly concerning severe disease and death [2]. Compared to children and adolescents, adults are more vulnerable to severe forms of COVID-19, particularly those with pre-existing comorbidities, older age, and other risk factors [3–6]. There has been an ongoing debate whether children could be a substantial, undetected source of SARS-CoV-2 transmission in the community given their milder disease and higher frequency of asymptomatic infections compared to adults [7]. A better understanding of their susceptibility and infectiousness among children helps understand their role in the dynamic of the COVID-19 pandemic.

As new variants of concern (VOCs) emerged worldwide, the epidemiological effects associated with their circulation also changed. Alpha and delta variants have been shown to be more transmissible than previous strains [8]. However, most studies have been performed among

adults, and it is unclear how viral evolution has affected susceptibility and infectiousness of children and adolescents. A common approach often used to assess the epidemiological properties of a given infectious agent is to measure transmission to household contacts after introduction of the agent through an index case [9]. In general, advantages of the household setting are that the transmission risk in households is high. Households can be considered mini cohorts where it is possible to study both susceptibility and infectiousness of children and adolescents, using adult household members as reference point. This study aimed to determine the relative SARS-CoV-2 susceptibility and infectiousness of SARS-CoV-2 in children (0–9 years old) and adolescents (10–19 years old) relative to adults, stratified by different viral variants (wild-type, alpha and delta), based on household contact data. We endeavoured to identify the effect of the variants themselves independent of the effect of vaccination or previous infection.

## Methods

### Protocol registration

This report was structured according to the Preferred Reporting Items for Systematic Reviews and Meta-Analyses (PRISMA) statement guidelines [10]. The protocol was registered with PROSPERO (CRD42021271023).

### Eligibility criteria

Studies were included if they met the following eligibility criteria: (A) they reported household secondary attack rates (SAR) or included the data required to compute household SAR with the possibility to compare children (0–9 years old), adolescents (10–19 years old), or both combined (0–19 years old) with adults (20+ years old); (B) they tested—at a minimum—all symptomatic household contacts for SARS-CoV-2 infection by reverse transcription polymerase chain reaction (RT-PCR); and (C) reported results regarding SARS-CoV-2 for wild-type (ancestral type), alpha or delta variants. For wild-type studies, we included studies that collected data before 2021, i.e., before VOCs alpha and delta began to circulate widely.

### Information sources and search strategy

We formulated a search strategy to identify all relevant studies regardless of language or publication status in EMBASE, MEDLINE, and medRxiv up to January 2022. We used the following index terms in PubMed: COVID, children, infectiousness, susceptibility and other terms paraphrasing the infectiousness and susceptibility concepts. Full search terms are included in Annex 1 in S1 File. A 'backwards' snowball search was conducted of the references in all included articles. Due to the likelihood of a high volume of relevant studies for the wild-type virus, we followed standard guidelines for integrating existing systematic reviews into new reviews [11, 12]. Where existing systematic reviews with equivalent search and study selection methods were available, these were used as a starting point to identify relevant studies.

### Study selection

Two reviewers independently screened titles and abstracts. We obtained full reports for all titles and abstracts that met the inclusion criteria or where there was any uncertainty. Full-text articles were reviewed to assess eligibility, and exclusion reasons were coded. Disagreements were resolved by consensus or by a third author if necessary.

## Data extraction and quality assessment

Citations extracted from electronic databases were imported to EndNote. The Covidence systematic review software [13] was used for the screening and review processes, including removal of duplicates. The reviewers independently used a pre-defined and piloted standardized tool to extract data pertinent to the research questions. Two reviewers independently extracted data from publications and reports and entered them in Microsoft Excel. Extracted data included all the necessary information to describe and characterise the studies, perform the quality assessments, synthesise data for the meta-analyses and assess heterogeneity. In case of missing data or insufficient reporting of details, the studies' corresponding authors were contacted for clarification or with a request to recalculate data to match age groups as used here. In addition, only data for unvaccinated persons were used or requested for studies on the delta variant (which circulated when COVID-19 vaccines were already broadly available). After data extraction was completed, codebooks were reviewed, and any discrepancies were resolved by consensus or by a third author if necessary.

The major categories of extracted data were: (1) study characteristics (author, journal, year of publication, country/region, funding sources); (2) study design (type of study, study period, duration of follow-up); (3) study population (sample size at baseline, population characteristics); and (4) case definition of index and household secondary case; (5) vaccination status of participants.

## Study risk of bias assessment

To assess the methodological quality and risk of bias of included studies, we used the modified version of the Newcastle-Ottawa quality assessment scale for observational studies [14]. Studies received as many as 9 points based on participant selection (4 points), study comparability (1 point), and outcome of interest (4 points). Studies were classified as having high ($\leq$3 points), moderate (4–6 points), and low ($\geq$7 points) risk of bias.

## Definitions, data synthesis and analysis

We defined a "household" [15] as a group of people, often a family, who live together in a house or flat. We referenced in eTable 1 in S1 File the terms "index case" or "primary case" as used in the studies themselves (**eTable 1 in S1 File**). While these terms have different meanings (index case is usually defined as the first case to be confirmed in a household, primary case is usually the case with the earliest date of symptom onset) for consistency purposes and to avoid confusion we have used the term "index case" for the rest of the paper. "Household contacts" (HHC) were defined as persons having been exposed to an index case and deemed susceptible (unvaccinated and–if available—with no history of SARS-CoV-2 infection). A "secondary case" was defined as a HHC with a positive RT-PCR test. The "household secondary attack rate (SAR)" was defined as the proportion of all HHC who were reported to have tested positive for SARS-CoV-2 divided by all susceptible HHC. We estimated **susceptibility** of an age group as the SAR of HHC of that age group, irrespective of the age of the index case. **Infectiousness** of an age group was the SAR of HHC (irrespective of age) exposed to index cases in that age group. As a relative measure for susceptibility, we estimated odds ratios (OR) comparing the SAR of children and/or adolescent HHC versus the SAR of adult HHC. As a relative measure for infectiousness, we compared SAR when children and/or adolescents were index cases versus when adults were index cases. We defined children as those persons aged 0–9 years, adolescents as persons aged 10 to 19 years, and adults as persons aged 20+ years. In three studies a slightly deviant age grouping was used which we accepted to be used for the three age groups children/adolescents/adults (Grijalva (<12, 12–17, 18+) [16]; Koureas (<12,

13–19, 20+) [17]; Singanayagam 2021 (4–9, 14–19, 20+) [18]). We avoided unit-of-analysis error that could be created by double-counting the participants in the multiple groups from one study, i.e. children versus adolescents versus adults, by splitting the "shared" adult group into two by dividing both the number of events (infected) and the total number of participants by two [19, 20]. To "isolate" the effect of the variants alone we extracted data on SAR among non-vaccinated household contacts infected from non-vaccinated index cases. Thus, if studies did not explicitly state that vaccination was an exclusion criterion, we felt it was reasonable during wild-type and alpha to assume that the index cases and household contacts were previously uninfected and unvaccinated (eTable 1 in **S1 File**).

We pooled ORs across studies using the DerSimonian and Laird random-effects model due to expected heterogeneity. We termed non-overlapping confidence intervals (CI) as statistically significant ORs. The $I^2$ statistic was used to evaluate heterogeneity between studies. We used an $I^2 > 50\%$ as indicating statistically significant heterogeneity. All statistical analyses were performed in R (version 3.6.1) with freely available statistical packages 'meta' (version 5.2.0) [21].

Data from Danish studies were of high quality. Results are therefore also presented separately for these data in the analysis. The data from these studies were collected consistently during circulation of all variants with the same methodology, had a large sample size, and provided data from unvaccinated index cases to unvaccinated household contacts, and included only households with no previously documented SARS—CoV-2 infection.

## Results

### Search results and study characteristics

Our systematic review search yielded 6,341 articles. **Fig 1** shows the study selection flow diagram. After deduplication and screening of articles using title and abstract, 218 full-text articles were selected for critical examination. Of these, 190 articles were further excluded because they failed to meet the inclusion criteria, largely, either because the results were based on non-household or mixed contacts (household/non-household not separated), or no relevant outcome data were reported. The remaining 27 articles [16–18, 22–45] involving 308,681 participants, met the inclusion criteria and were included in the review. Of the 27 studies, 20 (71%) were published in peer-reviewed journals, and 7 were accessible as pre-prints. The studies were based on surveillance data (n = 12, 44%), cohort study data (n = 11, 41%), register data (n = 3, 11%), and one retrospective case series (n = 1, 4%). Most of the studies were from China (n = 8, 30%), followed by the United States of America (n = 6, 22%). The majority of included studies reported on the wild-type virus (n = 21, 78%), followed by alpha variant (n = 3, 11%), wild-type and alpha (n = 2, 7%), delta (n = 1, 4%), and wild-type, alpha and delta (n = 1, 4%). The **S1 File** shows detailed information on data extracted from each study in **eTable 1 in S1 File**. The risk of bias in included studies is shown in **eTable 2 in S1 File**. The study quality total rating score ranged from 5 to 9 of a total of 9 possible points. Most of the studies were rated as having a low risk of bias (n = 22, 81%) and five studies were rated as having a moderate risk of bias (18%).

### Susceptibility to infection

**Wild-type virus.** Almost all the studies reported results for the wild-type virus (23 of 27 studies). The reported SAR varied greatly across studies, ranging from 0% to 56% in children, 5% to 47% in adolescents and 12% to 66% in adults (**eFigure 1 in S1 File**). Moreover, the number of contacts (denominator) varied from 2 (in the children subgroup) to 1,934 across studies, implying large differences in the precision of the studies' estimates. Heterogeneity of the SAR

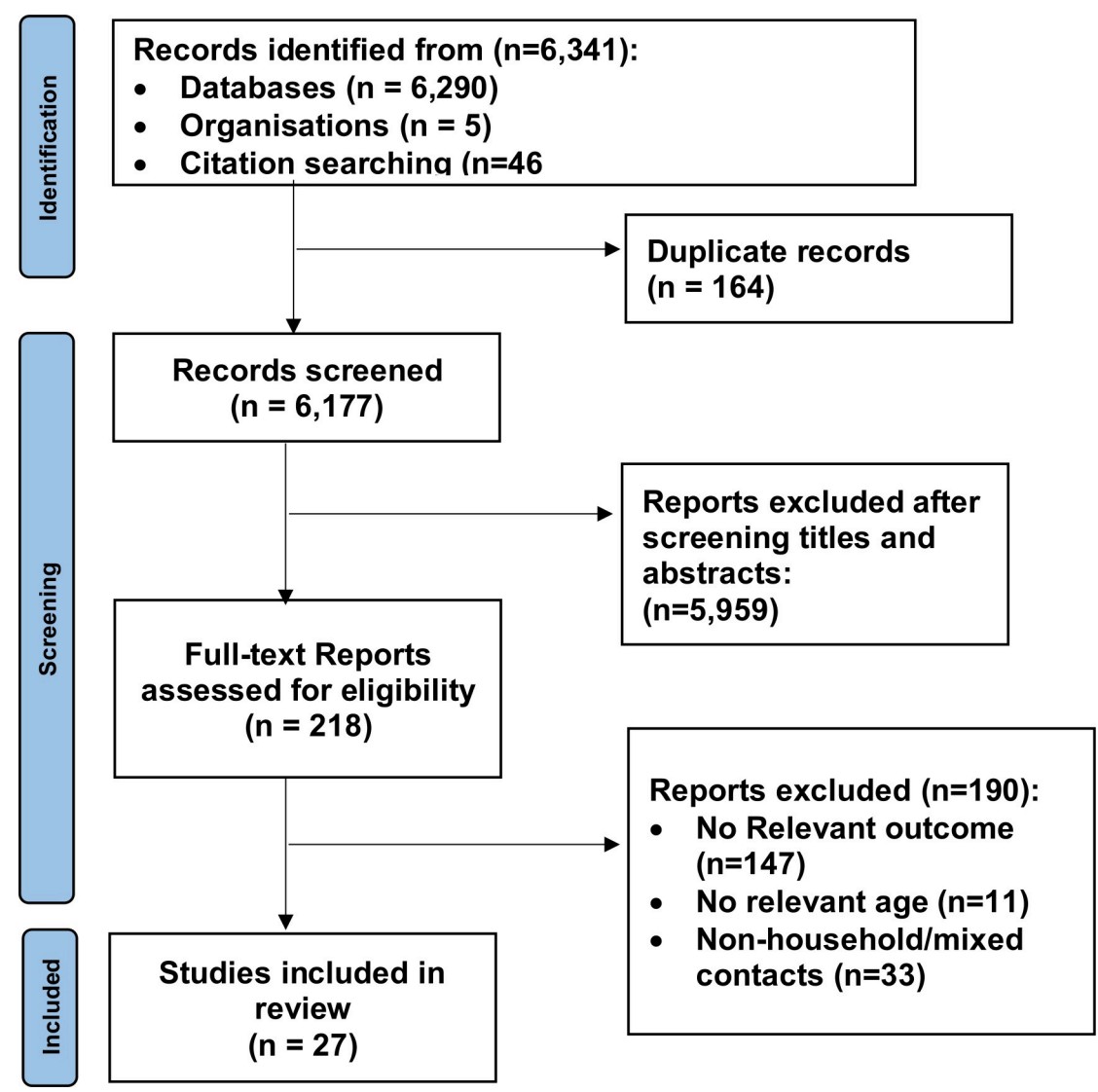

**Fig 1. PRISMA flow for study selection.**

was 94%, 87% and 99% (**eFigure 1 in S1 File**), respectively, for the three age groups. The heterogeneity of the OR for SARS-CoV-2 infection comparing children to adults and adolescents to adults was smaller than the heterogeneity for the SAR (42% and 0%, respectively; **Fig 2**). Compared to adults, the OR for SARS-CoV-2 infection in children was 28% lower, although not statistically significant (OR = 0.72, 95% CI 0.49–1.05, nine studies), whereas Danish data yielded an OR of 0.84 (95% CI 0.73–0.96) showing statistical significance [43, 46] (**Figs 2, 3**). Adolescents were 23% less susceptible (OR = 0.77, 95% CI 0.68–0.88, nine studies) than adults (**Figs 2 and 3**). Both the pooled estimate derived from the studies reporting on the 0–19 years old age group (OR = 0.43; 95% CI 0.24–0.75, 10 studies) as well as the estimate pooled for children and adolescents (OR = 0.58, 95% CI 0.44–0.77, 19 studies) indicated statistical significance.

**Alpha variant.** We identified six studies reporting results for this variant [18, 31, 35, 36, 38, 45]. The reported SAR for households infected with alpha was higher than for households

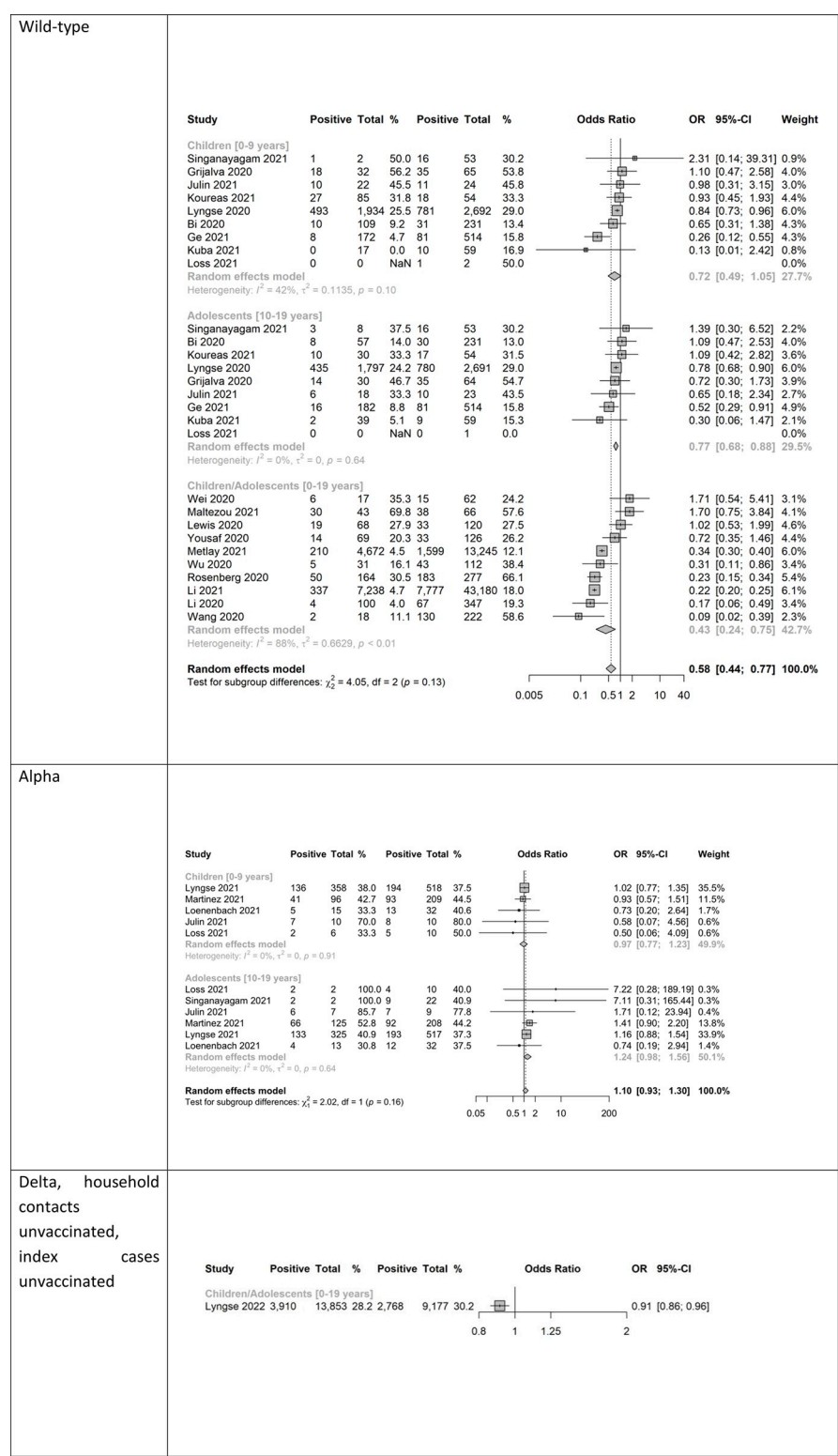

**Fig 2. Susceptibility analysis showing forest plot comparing odds ratios (OR) of secondary attack rates among (0–9 years old) child contacts and adults, among (10–19 years old) adolescents and adults, as well as among children and adolescents combined (0-19-year-old) and adults; wild-type, alpha variant, delta variant; ordered by descending OR.** Note: NaN—Not a number, inestimable, mean (0/0).

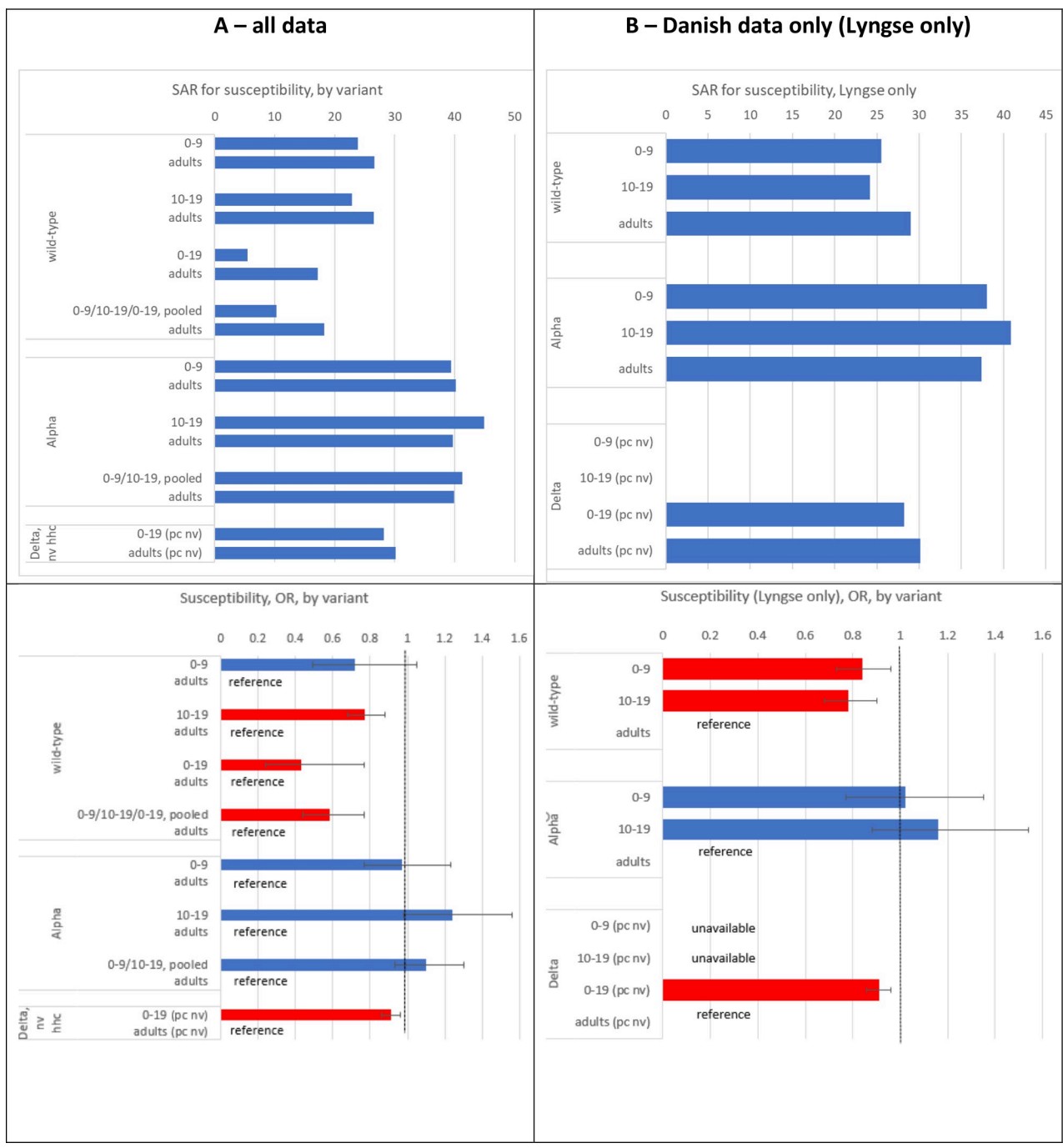

**Fig 3.** Susceptibility analysis: Secondary attack rates (SAR) of household contacts of children, adolescents and adults (top) as well as odd ratios (OR) (bottom) for SARS-CoV-2-infection among children and adolescents when compared to adults; wild-type, alpha variant, delta variant. To better visualize the point estimates of SAR, 95% confidence intervals are omitted. A (left): all included studies, B (right): Danish studies only [46, 70, 71]. Red bars are statistically significant from 1 on a 95% level. Note: HHC: household contact, IC: index case, NV: non-vaccinated, V: vaccinated.

infected with the wild-type but varied across studies and ranged from 33% to 70% in children, 31% to 100% in adolescents and 37% to 79% in adults (**eFigure 2 in S1 File**). Heterogeneity of SAR (13%, 69% and 73% among children, adolescents and adults, respectively; **eFigure 2 in S1 File**) was less than in wild-type studies (94%, 87%, 99%, respectively; **eFigure 1 in S1 File**), but

higher than for the OR of the comparisons children-adults (0%) and adolescents-adults (0%; Fig 2). Compared to adults, with point estimates of the OR close to 1 and above 1, comparisons in both age groups were not statistically significant (children vs. adults: OR = 0.97, 95% CI 0.77–1.23; five studies; adolescents vs. adults: OR = 1.24, 95% CI 0.98–1.56, six studies) (**Fig 2**). There were no additional studies reporting on the age group 0–19 years old. The pooled estimate of the studies used for the two comparisons above was not statistically significant from 1 (OR = 1.10, 95% CI 0.93 to 1.30).

**Delta variant.**   We only identified one large study from Denmark [44]. Compared to adults, children (0–19 years) were 9% less susceptible (OR = 0.91, 95% CI 0.86 to 0.96) (Fig 2) [44]. Overall, the crude SAR increased from about 5–25% for the wild-type virus to 40–45% for the alpha variant and dropped approximately to 30% in the delta variant (**Fig 3**). The OR increased from below 1 for the child age groups during wild-type circulation to around 1 during alpha and slightly (but significantly) below 1 during delta circulation (**Fig 3**).

## Infectiousness

**Wild-type virus.**   The reported SAR varied greatly across studies, ranging from 0% to 53% in children, 7% to 50% in adolescents and 12% to 57% in adults (**Fig 4**). In 6 of 7 studies the SAR of household contacts was lower when children were the index case (versus when adults were the index case), and the overall OR was 0.85, although not statistically significant from 1 (95% CI 0.70–1.03; eight studies). Also, the overall OR estimate for adolescents (versus adults) was not statistically different from 1 (OR = 0.82, 95% CI 0.54–1.26, nine studies) (**Figs 4, 5**). The estimate pooled for all studies and both age groups (children, adolescents) showed that children/adolescents/combined were 30% less infectious than adults (OR = 0.70, 95% CI 0.52–0.94, 13 studies) (**Fig 4**). The Danish study alone indicated statistically significantly lower infectiousness when children or adolescents were the index case (OR = 0.71, 95% CI 0.67–0.75; not displayed) [43].

**Alpha variant.**   In both age group comparisons (children vs adults, adolescents vs adults), five studies contributed data, and in both age group comparisons only two studies contributed more than 100 participants. The reported SAR varied a lot across studies, ranging from 39% to 100% in children, 0% to 100% in adolescents and 0% to 81% in adults (**Fig 4**). The children-adult comparison estimated that children were 35% more infectious than adults (OR = 1.35, 95% CI 1.01–1.79, five studies). There was no statistically significant difference in the OR estimates when adolescents (versus adults) were index cases (OR = 1.06, 95% CI 0.23–4.83, five studies) (**Figs 4 and 5**). Also, the pooled estimate for children and adolescents (versus adults) was not statistically significantly different from 1 (OR = 1.05, 95% CI 0.66–1.66).

**Delta variant.**   We only identified one large study from Denmark [44]. Compared to adults, children (0–19 years) were 29% less susceptible (OR = 0.71, 95% CI 0.67 to 0.75) (**Fig 4**) [44].

## Discussion

Our systematic review and meta-analysis provide an assessment of SARS-CoV-2 susceptibility and infectiousness in children (0–9 years old) and adolescents (10–19 years old) compared to adults (20+ years old) derived from data in household settings stratified by the dominating circulating strains wild-type, alpha, and delta. Children and adolescents were less susceptible to the wild-type and likely the delta variant but equally susceptible to the alpha variant as adults. Similarly, children and adolescents infected with the wild-type were less infectious than adults and likely less infectious when infected with the delta variant. However, alpha variant-related infectiousness is unclear; possibly 0–9 year old children were even more infectious than adults.

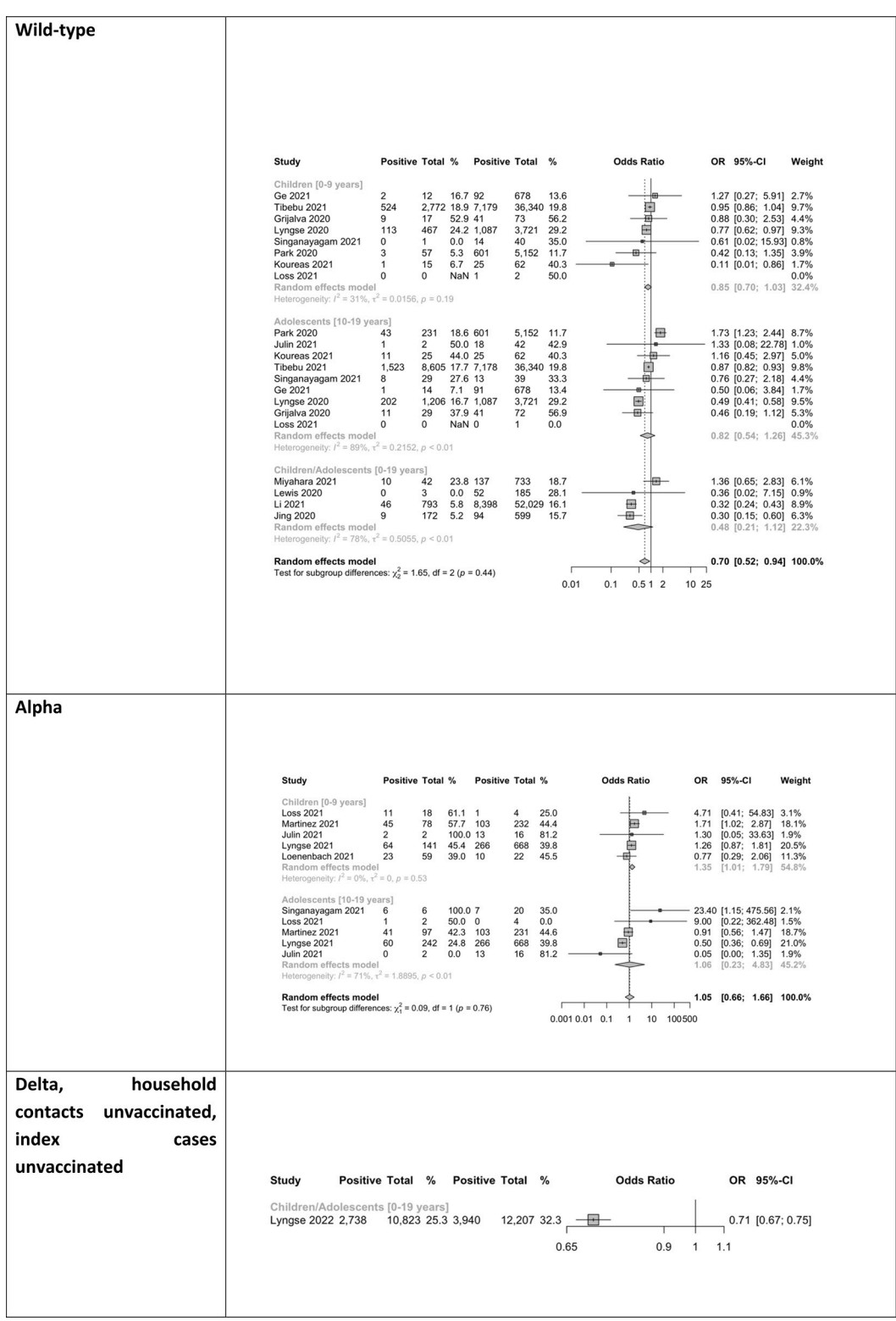

**Fig 4. Infectiousness analysis showing forest plot comparing odds ratios (OR) of secondary attack rates among household contacts when (0–9-year-old) children or adults were index cases, when (10–19-year-old) adolescents or adults were index cases, as well as when children or adolescents (0-19-year-old) were index cases; wild-type, alpha variant, delta variant; ordered by descending OR.** Note: NaN—Not a number, inestimable, mean (0/0).

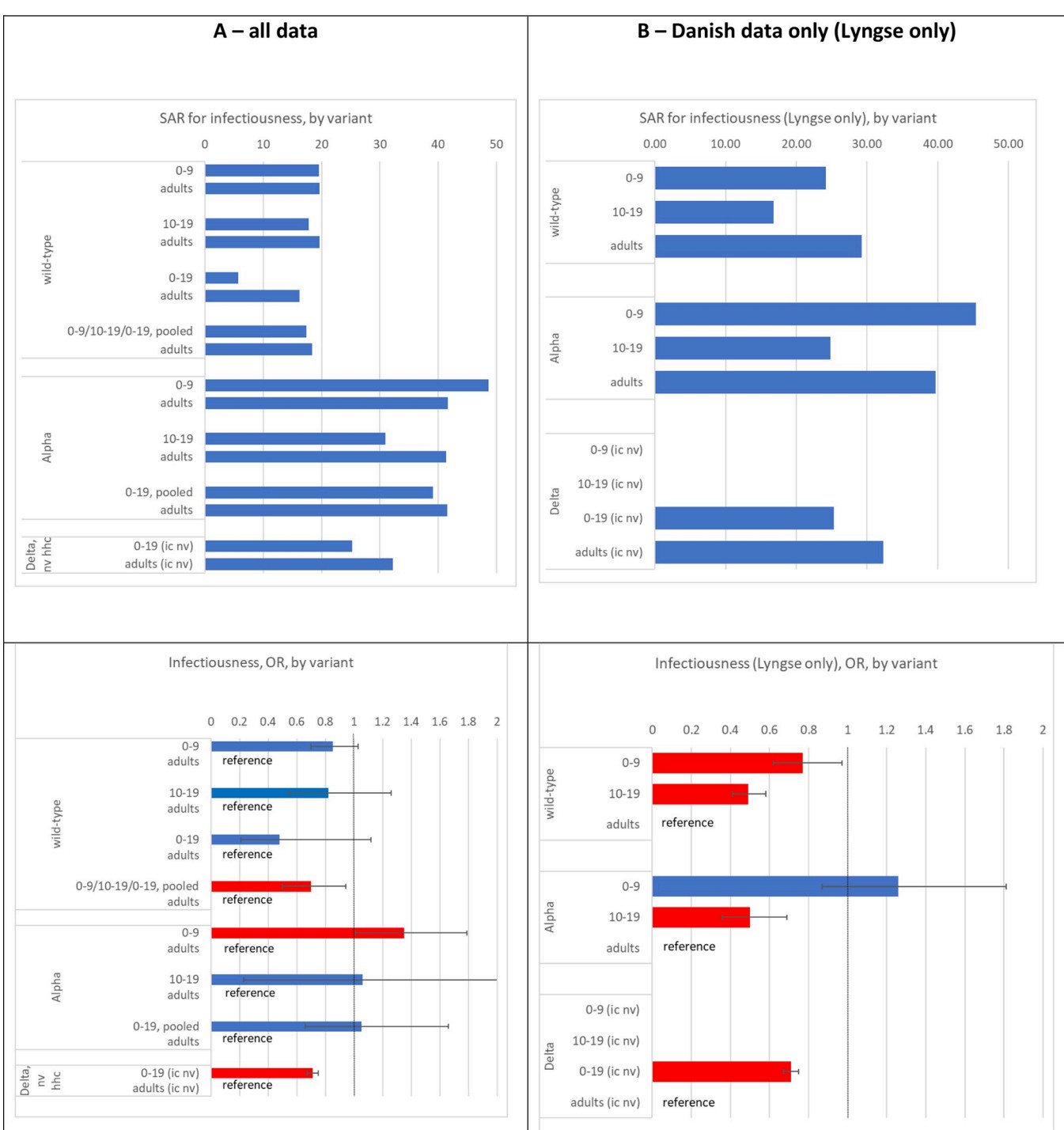

**Fig 5.** Infectiousness analysis: Secondary attack rates of household contacts when children, adolescents and adults were index cases (top), as well as odds ratios (OR) for SARS-CoV-2-infection when children vs adults, adolescents vs adults, or combined 0–19 year olds vs adults were index cases; wild-type, alpha variant, delta variant (bottom). To better visualize the point estimates of SAR 95% confidence intervals are omitted. A (left): all included studies, B (right): Danish study only [70]. Red bars are statistically significant. Note: HHC: household contact, IC: index case, NV: non-vaccinated, V: vaccinated.

The crude SAR among household contacts was lowest during wild-type circulation, highest during circulation of the alpha variant, and intermediate during delta circulation.

This type of information is generally valuable because households are an important "gear changer" where infections may be lifted or moved down from one age group level (e.g. children) to another (adolescents, or parents in working age). Thus, it adds information on the preferred directionality. For example, children often acquire influenza infection from peers, introduce them into their households and their parents promote them in their own age group, e.g. through the means of work contacts. This type of (age stratified) information is particularly important at the beginning of a pandemic, because, if children and adolescents are more on the "receiving side", closures of schools and/or daycare centers with the aim to contain spread in the community might be scrutinized.

Our finding that children of 0 to 9 years and adolescents of 10 to 19 years were less susceptible to the wild-type strain of SARS-CoV-2 than adults, is consistent with the comprehensive review and meta-analysis of household studies by Madewell et al. [47] and Zhu [48] who, however, did not differentiate between children and adolescents. Irfan [49] and Viner [50] (covering studies during wild-type circulation) disaggregated into children and adolescents and agreed in that the odds of infection in children is significantly less than for adults, but stated that adolescents are as susceptible or even more susceptible than adults. However, in contrast to our approach, both studies included not only the household, but also other settings, which may explain the difference to our results.

Several hypotheses have been proposed why during circulation of the wild-type SAR in children were found to be lower than that in adults, such as lower probability of testing, lower viral load [51], innate immunity [52, 53] or cross protection through cellular immunity from previous exposure to seasonal corona viruses [54, 55]. At any rate, lower susceptibility of children compared to adults stands in contrast to influenza, where several household studies have found that children (mostly pooled in broad age groups, e.g. <18 years) are more susceptible than adults [56–59]. However, not all child age groups may be the same. When broken down into smaller age groups one study conducted before the influenza pandemic (A/H1N1) 2009 and another conducted during the (A/H1N1) 2009—pandemic found that children younger than 6 and 5 years, respectively, may be as susceptible as adults [58, 60]. This fact also motivated us to differentiate at least between two child age groups.

The situation for the variants following the wild-type is less clear. Circulation of the alpha variant generally seems to have been associated with a substantially higher SAR among both children and adolescents similar to that among adults. Evaluation of the delta variant is complicated by increasing vaccination coverage and prior infection. Only the Danish data were able to analyse effects of the delta variant alone, i. e. when exposure occurred in households without prior infection, and from unvaccinated index cases to unvaccinated household contacts. Results suggest that at least for the combined age group of 0–19 year olds susceptibility is again lower compared to adults. After omicron and its variants or recombinants began to dominate worldwide such studies have become more difficult to interpret as the vast majority of the population either has been vaccinated various times or had experienced at least one, not always documented, SARS-CoV-2 infection [61].

Our analysis suggests that susceptibility and infectiousness of child age and adolescent age groups in relation to adults has changed from variant to variant. However, it has not changed in the same direction and magnitude. Thus, susceptibility and infectiousness may have to be reassessed with every new variant anew. Although after the advent of VOC omicron analyses restricting to unvaccinated index cases and unvaccinated (and previously uninfected) household contact persons have almost become impossible further household transmission studies are warranted nevertheless but will have somewhat different interpretations compared to this study.

## Study strengths and limitations

The main strength of our review is the strict methodological process. We followed Cochrane Collaboration, PRISMA, and GRADE recommendations, performed two comprehensive searches, and used duplicate data extraction and risk of bias assessments by two independent authors. Most of the included studies were at low risk of bias. Compared with previous reviews [48–51, 62–65], we sought to further characterize the risk of both susceptibility and infectiousness and we have undertaken pre-specified sub-group analyses by age group (by children as well as by adolescents) and variant types. Furthermore, to improve comparability, we have restricted our analysis to household contacts alone, while many other studies mix in data from contact-tracing activities or do not differentiate between household and non-household contacts. By analysing the susceptibility and infectiousness relative to adults, we automatically were able to adjust for unknown conditions that may influence the SAR within households in various ways, such as cultural behaviour or phase of the pandemic. For example, the estimates of SAR of susceptibility for wild-type variants were very heterogeneous for both children, adolescents, and adults (eFigure 1 in **S1 File**), whereas the OR estimates were more homogenous (Fig 2). It is also a strength that one of the authors (FL) contributed large data with the same study design (register study) and methodology for all three variants. In addition to providing precise estimates from a large number of households, using the same methods across time allows for comparisons, while holding possible confounding conditions constant.

There are several limitations to our study. If household transmission is not closely monitored the true primary case may not be the index case. As children/adolescents (0–19 year old) are more frequently asymptomatic than adults [66], it is possible that children (and adolescents) are partially missed as primary cases and the SAR associated with child index cases is underestimated when compared to adult index cases [67]. Results of a childcare outbreak/household study in Germany corroborate this consideration [36].

As the majority of studies were conducted in high- or middle-income countries, with 8 studies (30%) from China and 6 studies (22%) from the United States. Household sizes in these settings may differ from those in low-income countries, potentially affecting secondary attack rates and other household properties.

We did not take the timing of the epidemic into account in this study (e.g., epidemic peak vs. decline). We concentrated on variant-specific analyses. Depending on the timing of each study, times of open or closed school or kindergartens (due to mandate or vacation) may affect contact time within the household and–vice versa–limit contacts outside.

We did not extend our analysis to the omicron variant mainly because of the dearth of studies. A few studies indicate that both children and adults are more susceptible and more infectious in regards to the omicron variant compared to the delta variant [68] which holds for both BA.1 as well as BA.2 [69]. However, this needs to be verified through further studies.

We did not examine differences in susceptibility and infectiousness between vaccinated and unvaccinated household contacts as our inclusion criteria and analyses were designed to isolate the effects of each SARS-CoV-2 variant independent of immunity from vaccination or prior infection. While this allowed us to assess the intrinsic properties of the variants themselves, it does not capture the real-world impact of vaccines that were widely available during circulation of the Delta variant and beyond. Future household transmission studies should consider stratifying by vaccination status or measure immunity status serologically, as understanding susceptibility and infectiousness differences between vaccinated and unvaccinated individuals is relevant for guiding public health policies and recommendations.

Another potential limitation is that we did not assess differences across studies in guidance on isolation of infected individuals to protect household members. Such guidance likely varied

between and within countries over time, and adherence within households may have been inconsistent and would have to be recorded also. Stricter isolation policies could reduce household transmission, while limitations in household space or resources could make effective isolation difficult. As long as household studies are not standardized this type of information will not be available in an analyzable way. Only randomized trials assigning households to different isolation protocols could provide more definitive evidence, but such designs may be logistically and ethically challenging. Our systematic review includes literature published up to January 2022, and as such only one study on the Delta variant met our inclusion criteria. It is possible that additional household transmission studies of the delta variant have been published. The single included (Danish) study may not fully capture the heterogeneity of delta transmission across different settings and populations. Having said that we believe that not many study groups will have been able to control for vaccination and previous infections and with the same statistical power as has been done in the cited Danish study which is included in this review.

The results of the pooled data should be interpreted with caution, as in any active outbreak response, there was high heterogeneity across studies in study design (e.g., follow-up duration, frequency of testing, and universal and/or symptomatic testing), age groups were not standardized, transmission mitigation strategies after index case diagnosis, follow-up, testing strategies, household crowding, underlying seroprevalence, and other factors may have differed. In particular, testing of children is prone to bias, as they may be less frequently tested. Retrospective register (surveillance) studies may underestimate SAR in general and especially for children.

As is customary for household studies, we assumed that secondary SARS-CoV-2 infections among household contacts of the index (assumed primary) case were the result of a direct transmission event. It is also possible that the index case in the family is not the primary case [48]. We were unable to control for the chance of a 'common exposure' where two individuals were infected by the same source at the same time, but one individual was incorrectly identified as the sole index case of the cluster as they were the first to develop symptoms [48]. Furthermore, concurrent community acquisition may be different for different age groups. Likewise, it is important to note that these data should not be extrapolated to SARS-CoV-2 transmission outside the home where older children and adolescents tend to have more social contacts than adults [3].

## Conclusion

Our results show that children and adolescents are both susceptible and infectious to SARS-CoV-2 and their role in overall transmissibility likely changed with new variants emerging. Children and adolescents were less susceptible to the wild-type and probably also the delta variant of SARS-Cov-2 infection, they were less infectious when infected with the wild-type and probably also when infected with the delta variant. For the alpha variant, data suggest that children and adolescents were as susceptible as adults, and that children were at least as infectious as adults. These results show the importance of understanding the relative transmissibility of children and adolescents and the heterogeneity across variants. In consequence, the relative susceptibility and infectiousness of children and adolescents may have to be reassessed for every new variant anew.

## Supporting information

**S1 Checklist. Preferred Reporting Items for Systematic reviews and Meta-Analyses (PRISMA) checklist.**
(DOCX)

**S1 Data. Data to replicate secondary household attack rates (SAR) for SARS-CoV-2 infection in children (0–9 years), adolescents (10–19 years) or both compared with adults (20 years and older).**
(XLSX)

**S1 File.**
(DOCX)

## Author Contributions

**Conceptualization:** Olalekan A. Uthman, Frederik Plesner Lyngse, Seun Anjorin, Barbara Hauer, Seran Hakki, Udo Buchholz.

**Data curation:** Olalekan A. Uthman, Frederik Plesner Lyngse, Seun Anjorin, Barbara Hauer, Udo Buchholz.

**Formal analysis:** Olalekan A. Uthman, Frederik Plesner Lyngse, Seun Anjorin, Udo Buchholz.

**Funding acquisition:** Olalekan A. Uthman, Seun Anjorin, Udo Buchholz.

**Investigation:** Olalekan A. Uthman, Frederik Plesner Lyngse, Seun Anjorin, Barbara Hauer, Diego A. Martinez, Yang Ge, Jakob Jonnerby, Cathinka Halle Julin, Gary Lin, Ajit Lalvani, Julika Loss, Kieran J. Madon, Leonardo Martinez, Lisbeth Meyer Næss, Kathleen R. Page, Diana Prieto, Anna Hayman Robertson, Ye Shen, Juliane Wurm, Udo Buchholz.

**Methodology:** Olalekan A. Uthman, Frederik Plesner Lyngse, Barbara Hauer, Seran Hakki, Diego A. Martinez, Yang Ge, Jakob Jonnerby, Cathinka Halle Julin, Gary Lin, Ajit Lalvani, Julika Loss, Kieran J. Madon, Leonardo Martinez, Lisbeth Meyer Næss, Kathleen R. Page, Diana Prieto, Anna Hayman Robertson, Ye Shen, Juliane Wurm, Udo Buchholz.

**Project administration:** Olalekan A. Uthman, Seun Anjorin.

**Resources:** Olalekan A. Uthman.

**Software:** Olalekan A. Uthman.

**Supervision:** Barbara Hauer.

**Visualization:** Udo Buchholz.

**Writing – original draft:** Olalekan A. Uthman, Frederik Plesner Lyngse, Seun Anjorin, Barbara Hauer, Seran Hakki, Diego A. Martinez, Yang Ge, Jakob Jonnerby, Cathinka Halle Julin, Gary Lin, Ajit Lalvani, Julika Loss, Kieran J. Madon, Leonardo Martinez, Lisbeth Meyer Næss, Kathleen R. Page, Diana Prieto, Anna Hayman Robertson, Ye Shen, Juliane Wurm, Udo Buchholz.

**Writing – review & editing:** Olalekan A. Uthman, Frederik Plesner Lyngse, Seun Anjorin, Barbara Hauer, Seran Hakki, Diego A. Martinez, Yang Ge, Jakob Jonnerby, Cathinka Halle Julin, Gary Lin, Ajit Lalvani, Julika Loss, Kieran J. Madon, Leonardo Martinez, Lisbeth Meyer Næss, Kathleen R. Page, Diana Prieto, Anna Hayman Robertson, Ye Shen, Juliane Wurm, Udo Buchholz.

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
