## [Decision Letter · Decision Letter 0]

24 Aug 2023

PONE-D-23-01471Susceptibility and infectiousness of SARS-CoV-2 in children versus adults, by variant (wild-type, Alpha, Delta): a systematic review and meta-analysis of household contact studiesPLOS ONE

Dear Dr. Uthman,

Thank you for submitting your manuscript to PLOS ONE. After careful consideration, we feel that it has merit but does not fully meet PLOS ONE’s publication criteria as it currently stands. Therefore, we invite you to submit a revised version of the manuscript that addresses the points raised during the review process.

We look forward to receiving your revised manuscript.

Kind regards,

Victor Daniel Miron

Academic Editor

PLOS ONE

Journal Requirements:

2. We noted that the database search of your systematic review was carried out January 2022. Please ensure that your search is up to date and any relevant studies published since January 2022 are included in your systematic review. Thank you for your attention to this request.

3. When you submit your revised manuscript, please update the Editorial Manager submission form to include the correct email for co-author Yang Ge.

This research was commissioned by the Robert Koch Institute, Germany (OAU, SA, UB, BH). 

**Editor Comments**:

Although the Alpha and Delta variants are no longer circulating today, your analysis is of interest as it provides insight into the impact that SARS-CoV-2 has had on its evolution. Throughout the pandemic, the pediatric population has been of particular interest to the medical world due to the response that children have had to the infection. The data you present is of interest and worth publishing.

I recommend a revision of the text to update it, given that it was written in 2022.

The introduction should be changed without reference to the number of cases and deaths.

Review the discussion on Omicron and the limitations part where it makes mention of Omicron.

Reviewers' comments:

Reviewer's Responses to Questions

**Comments to the Author**

1. Is the manuscript technically sound, and do the data support the conclusions?

Reviewer #1: Yes

2. Has the statistical analysis been performed appropriately and rigorously? 

Reviewer #1: Yes

3. Have the authors made all data underlying the findings in their manuscript fully available?

Reviewer #1: Yes

4. Is the manuscript presented in an intelligible fashion and written in standard English?

Reviewer #1: Yes

5. Review Comments to the Author

Reviewer #1: The authors have performed a a systematic reivew of susceptiblity and infectiousness of SARS-CoV-2 in children versus adults by variant type - wild type, alpha and delta.

The authors defined "household", "index/primary case", "household contact" NB acronym hhc should be capitalised.

Why should vaccinated individuals be excluded as a contact? Very relevant in delta (and more recent variants); and useful to understand the differences in susceptiblity in vaccinated versus unvaccinated individuals.

The authors mention that a large proportion of studies were from China and the USA, but would recommend mentioning whether all studies were performed in high-moderate income countries. As this is a systematic review, the authors are behest to the quality of the studies. That said, it is important for the reader to understand and would recommend the authors to comment on 1. average household size, 2. guidance on isolation away from household members (or note it as a limitation if unknown) as this will affect both susceptibility and infectiousness.

The authors mention that the search was conducted up to and including January 2022, and therefore only 1 delta household study has been included. I think the authors should acknowledge that further studies have been published, in other settings to better inform HH transmission of the delta variant.

Variants are not usually capitalised (e.g. should read alpha, delta, omicron etc)

Page 16 - consider revising "After the omicron such studies have..." (remove 'the')

6. PLOS authors have the option to publish the peer review history of their article (what does this mean?). If published, this will include your full peer review and any attached files.

Reviewer #1: No

---

## [Author Response · Author response to Decision Letter 0]

12 Jun 2024

Dear Editor,

I can confirm that “The funders had no role in study design, data collection and analysis, decision to publish, or preparation of the manuscript”

Thank you for considering our manuscript "Susceptibility and infectiousness of SARS-CoV-2 in children versus adults, by variant (wild-type, alpha, delta): a systematic review and meta-analysis of household contact studies" for publication in PLOS ONE. We appreciate the thoughtful comments from the reviewers and the opportunity to submit a revised version of our work.

As agreed with you we have not rerun the entire search as it would have resulted in too many new publications. We have, however, done the following:

1. We updated all preprint references to peer-reviewed publications where available and removed one study that had significant methodological changes between preprint and publication, which we considered inappropriate (for details see table with point-to-point reply).

2. we removed global specific case and death numbers from the Introduction.

3. we acknowledged the limited number of included delta studies as a limitation and noted that more recent data are perhaps available, although we doubt that they will be of a similar quality as the Danish study that is included in this review. In addition, please note that the objective of this study was to analyze the effect of the variants themselves, i.e. among immune-naïve persons. Since the advent of omicron, its variants and recombinants the same approach is virtually not feasible anymore.

4. Minor editorial changes for consistency and clarity were made (see track changes in revised manuscript).

With these amendments, we believe our work remains a valuable contribution to understanding the relative susceptibility and infectiousness of (immune-naïve) children and adolescents across SARS-CoV-2 variants. Our systematic approach focusing on household studies only and on isolating the effects of each variant independent of vaccination provide a rigorous analysis of this important public health question.

Below, we have provided detailed point-by-point responses to each of the reviewers' comments, outlining the specific changes made or justifications provided for our approach. We hope these will satisfactorily address any outstanding concerns.

Yours sincerely,

Ola Uthman for the Authors

---

## [Editor Report · Decision Letter 1]

24 Jun 2024

Susceptibility and infectiousness of SARS-CoV-2 in children versus adults, by variant (wild-type, alpha, delta): a systematic review and meta-analysis of household contact studies

PONE-D-23-01471R1

Dear Dr. Uthman,

We’re pleased to inform you that your manuscript has been judged scientifically suitable for publication and will be formally accepted for publication once it meets all outstanding technical requirements.

Kind regards,

Victor Daniel Miron

Academic Editor

PLOS ONE

---

## [Editor Report · Acceptance letter]

19 Jul 2024

PONE-D-23-01471R1 

PLOS ONE

Dear Dr. Uthman, 

I'm pleased to inform you that your manuscript has been deemed suitable for publication in PLOS ONE. Congratulations! Your manuscript is now being handed over to our production team.

Kind regards, 

on behalf of

Dr. Victor Daniel Miron 

Academic Editor

PLOS ONE